# The role of novel biomarkers in the early diagnosis of pancreatic cancer: A systematic review and meta-analysis

Zeyi Zheng[1], Ziyu Lu[2], Fei Yan[3], Yani Song[4*]

**1** School of Traditional Chinese Medicine, Inner Mongolia Medical University, Hohhot, China, **2** School of Basic Medicine, Inner Mongolia Medical University, Hohhot, China, **3** Department of Cardiovascular Surgery, Shandong Provincial Hospital Affiliated to Shandong First Medical University, Jinan, Shandong, China, **4** School of Water Resources and Hydropower Engineering, Wuhan University, Wuhan, Hubei, China

* ze96yi@163.com

## Abstract

### Background

Early detection of pancreatic cancer is essential for improving survival rates. However, noninvasive diagnostic methods are lacking. Novel biomarkers, detectable through liquid biopsy, such as circulating tumor DNA (ctDNA), microRNAs (miRNAs), protein markers, and metabolites, hold promise for early diagnosis.

### Methods

A systematic search of PubMed, Embase, Web of Science, and the Cochrane Library was conducted for studies published from January 2014 to May 2024. Studies were included if they evaluated novel biomarkers for early pancreatic cancer detection, reported diagnostic performance metrics (sensitivity, specificity), and had a QUADAS-2 score of ≥3. Data on study characteristics, patient demographics, biomarker types, and diagnostic performance were extracted following PRISMA guidelines. A bivariate random-effects model was used to calculate pooled sensitivity, specificity, positive likelihood ratio (PLR), negative likelihood ratio (NLR), and diagnostic odds ratio (DOR). The area under the summary receiver operating characteristic (SROC) curve assessed overall diagnostic accuracy. The primary outcome was the diagnostic accuracy (sensitivity and specificity) of novel biomarkers in detecting early-stage pancreatic cancer.

### Results

A total of 43 studies involving 19,326 participants were included, with 2,749 patients having stage I or II pancreatic cancer. The pooled sensitivities and specificities were as follows:.

**Data availability statement:** All relevant data are within the paper and its Supporting Information files.

**Funding:** The author(s) received no specific funding for this work.

**Competing interests:** The authors have declared that no competing interests exist.

miRNA Biomarkers: Sensitivity 0.88 (95% CI 0.79-0.93), Specificity 0.91 (95% CI 0.82-0.95), DOR 72.68 (95% CI 26.64-198.24), AUC 0.95.

Protein Biomarkers: Sensitivity 0.79 (95% CI 0.70-0.86), Specificity 0.88 (95% CI 0.82-0.93), DOR 27.74 (95% CI 14.32-53.76), AUC 0.90.

Metabolite Biomarkers: Sensitivity 0.84 (95% CI 0.73-0.92), Specificity 0.85 (95% CI 0.81-0.88), DOR 31.76 (95% CI 12.38-81.48), AUC 0.90.

ctDNA Biomarkers: Sensitivity 0.65 (95% CI 0.48-0.81), Specificity 0.94 (95% CI 0.88-0.97), DOR 27.73 (95% CI 12.91-59.55), AUC 0.92.

Subgroup analyses showed combining biomarkers with CA19–9 improved diagnostic accuracy. Sensitivity analyses confirmed the robustness of the findings.

## Conclusions

Novel biomarkers, particularly miRNAs and protein markers, demonstrate high diagnostic accuracy for early pancreatic cancer detection and have potential for clinical application in improving early diagnosis and patient outcomes.

## Systematic review registration

https://www.crd.york.ac.uk/prospero/, Identifier: PROSPERO (CRD42024553633).

## Introduction

Pancreatic cancer is among the most lethal malignancies, exhibiting the lowest survival rates across all cancer types [1]. It is projected to become the second leading cause of cancer-related deaths by 2030 [2]. This dismal mortality rate is primarily attributable to late-stage diagnoses, at which point curative treatments are often unfeasible. Early-stage surgical resection remains the principal potentially curative intervention for pancreatic cancer [3]. However, early-stage disease is frequently asymptomatic or presents with nonspecific symptoms such as fatigue, indigestion, and changes in bowel habits [4], complicating early detection efforts. One of the greatest challenges in the management of PDAC is achieving early detection in high-risk individuals and accurately diagnosing patients who present with suspected symptoms [5]. Consequently, over two-thirds of patients present with regional or distant metastases at diagnosis [6]. The five-year survival rate for patients diagnosed with localized pancreatic cancer is significantly higher than for those with advanced disease stages [7]. Therefore, early detection and diagnosis are critical for improving survival outcomes in pancreatic cancer.

Currently, effective methods for early detection are lacking. Imaging modalities such as computed tomography (CT), magnetic resonance imaging (MRI), and endoscopic ultrasound-guided fine-needle aspiration are limited by cost, invasiveness, and suboptimal sensitivity and specificity, especially in early-stage disease [8]. The most widely used noninvasive biomarker, serum carbohydrate antigen 19–9 (CA19–9), has limited sensitivity and specificity for early pancreatic cancer detection [9,10]. Elevated CA19–9 levels can occur in benign conditions like pancreatitis

and other malignancies, leading to false positives [11]. Additionally, approximately 5% to 10% of the population are Lewis antigen-negative and do not produce CA19–9, resulting in false negatives [12]. Research and application of novel biomarkers for the early detection of potentially curable pancreatic cancer remain in their infancy. There is an urgent need for accurate, cost-effective, and efficient noninvasive detection methods capable of diagnosing pancreatic cancer at a resectable stage, which is critical for improving the five-year survival rate of patients with PDAC [13].

Recent advancements have identified novel biomarkers detectable through liquid biopsies, such as circulating tumor DNA (ctDNA), microRNAs (miRNAs), specific protein markers, and metabolite biomarkers [14,15], which show promise in the early diagnosis of pancreatic cancer. These biomarkers offer the potential for noninvasive, sensitive, and specific detection methods. This meta-analysis aims to evaluate the sensitivity and specificity of various novel biomarkers in the early diagnosis of pancreatic cancer, compare their diagnostic performance, and identify the most promising biomarkers for clinical application.

## Methods

### Protocol and registration

We conducted this systematic review and meta-analysis following the Preferred Reporting Items for Systematic Reviews and Meta-Analyses (PRISMA) guidelines [16]. The study protocol was registered with PROSPERO (registration number CRD42024553633).

### Eligibility criteria

We included studies that met the following criteria:

1.  Study Design: Cohort studies, case-control studies, and diagnostic test accuracy studies.

2. Participants: Patients suspected of or diagnosed with pancreatic cancer.

3. Biomarkers: Evaluations of novel biomarkers such as ctDNA, miRNAs, or specific protein markers for early diagnosis.

4. Outcomes: Reports on diagnostic performance metrics, including sensitivity, specificity, positive predictive value (PPV), and negative predictive value (NPV).

5. Quality Assessment: Studies assessed using the Quality Assessment of Diagnostic Accuracy Studies-2 (QUADAS-2) tool were included only if they had a rating of "high risk" in at most one domain of bias [17].

Exclusion criteria were:

1. Non-human studies.

2. Studies lacking complete diagnostic performance data.

3. Duplicate publications.

4. Studies with excessive heterogeneity in design, patient population, or biomarker detection methods, as identified by QUADAS-2 evaluation.

5. Small-scale studies with fewer than 30 participants or conducted exclusively in specific regions or healthcare systems.

Exclusion of Pancreatic Cancer
In the included studies, patients with confirmed pancreatic cancer were excluded based on several criteria. First, any study focusing on patients already diagnosed with pancreatic cancer was excluded, with the diagnosis confirmed through clinical evaluation and medical records. Second, studies that included patients with a history of pancreatic cancer or evidence of malignancy, as determined by clinical assessment, imaging tests (such as CT or MRI), or biopsy, were also

excluded. Additionally, studies specifically targeting early-stage pancreatic cancer detection (e.g., stage I and II) excluded patients with advanced-stage cancer (III or IV), which was determined through clinical staging, imaging, and histopathological examination. These procedures were implemented to ensure that all included studies focused on identifying biomarkers for the early detection of pancreatic cancer, while excluding patients with a confirmed diagnosis of the disease.

## Search strategy

We performed a comprehensive literature search for studies published from January 2014 to May 2024. The search was conducted on June 1, 2024. Databases searched included PubMed, Embase, Web of Science, and the Cochrane Library. We combined MeSH terms and free-text keywords related to "pancreatic cancer," "early diagnosis," and "novel biomarkers" using Boolean operators (AND, OR).The specific search terms were:

Population: "pancreatic cancer" OR "pancreatic carcinoma"

Intervention: "biomarker" OR "circulating tumor DNA" OR "ctDNA" OR "microRNA" OR "miRNA" OR "protein marker"

Outcome: "early diagnosis" OR "early detection" AND "diagnostic accuracy" OR "sensitivity" OR "specificity"

We also screened reference lists of relevant articles to identify additional studies.
The specific search strategy is detailed in the attached table (S1 Table).

## Study selection

The search strategy results were stored and managed using the Endnote software. Two independent reviewers screened the titles and abstracts of all retrieved articles. Full-text articles were obtained for studies meeting the inclusion criteria or when eligibility was uncertain. Discrepancies were resolved through discussion or consultation with a third reviewer.

## Data extraction

Two reviewers independently extracted data using a standardized form, including:

Study Characteristics: Author, publication year, country, study design, and sample size.

Patient Characteristics: Age, sex, and disease stage.

Biomarker Details: Specific biomarkers evaluated and detection methods used (e.g., PCR, NGS).

Diagnostic Performance Metrics: Sensitivity, specificity, PPV, NPV, and 95% confidence intervals.

Quality Assessment: QUADAS-2 scores for patient selection, index test, reference standard, and flow and timing.

Any discrepancies were resolved by consensus or involving a third reviewer.
Missing data were managed by initially contacting the corresponding authors of the original studies to obtain any unavailable or unclear information. In the absence of a response, missing values were estimated from figures or derived through appropriate statistical conversions based on the available data. Studies with substantial missing data that could not be reliably imputed were excluded from the meta-analysis.

## Quality assessment

We assessed the methodological quality of the included studies using the QUADAS-2 tool, which evaluates the risk of bias across four domains (patient selection, index test, reference standard, flow and timing) and applicability concerns across three domains (patient selection, index test, reference standard). Two reviewers independently rated the risk of bias and applicability concerns for each domain as "low," "high," or "unclear." Studies were included if at most one bias

domain was rated as "high risk" (out of four bias domains). Applicability concerns were documented but did not influence the inclusion decision, as they pertain to generalizability rather than internal validity. Disagreements between reviewers were resolved through discussion or consultation with a third reviewer.

### Statistical analysis

We used Stata version 16 for statistical analyses. A bivariate random-effects model calculated pooled estimates of sensitivity, specificity, positive likelihood ratio (PLR), negative likelihood ratio (NLR), and diagnostic odds ratio (DOR) with 95% confidence intervals. We constructed summary receiver operating characteristic (SROC) curves and calculated the area under the curve (AUC) to evaluate overall diagnostic accuracy.

Heterogeneity among studies was assessed using Cochran's Q test and the $I^2$ statistic. An $I^2$ value over 50% indicated substantial heterogeneity [18,19]. We explored potential sources of heterogeneity through subgroup analyses based on biomarker type, sample type (blood, urine), and detection methods (e.g., PCR, NGS).

Sensitivity analyses excluded studies with a high risk of bias to assess the robustness of pooled estimates. We evaluated publication bias using funnel plots and Deeks' funnel plot asymmetry test.

### Results reporting

We reported results following PRISMA guidelines. A flow diagram illustrated the study selection process. Tables summarized the characteristics of included studies and quality assessment outcomes. Forest plots presented pooled diagnostic performance metrics. We documented assessments of heterogeneity and publication bias and reported findings from subgroup and sensitivity analyses.

## Results

### Study selection and characteristics

Our search yielded 1,742 studies, with 502 duplicates removed. After screening titles and abstracts, we selected 222 studies for full-text evaluation. Of these, 178 were excluded for reasons such as not being diagnostic tests, unclear cancer stage definition, lacking specific biomarkers, incomplete diagnostic data, and sample sizes of fewer than 30 participants. An additional two studies were excluded due to insufficient numbers for meta-analysis. Ultimately, 42 studies met the eligibility criteria (Fig 1).

Tables 1 and 2 summarize the included studies, published between 2014 and 2024, involving 19,326 participants. Among them, 2,749 patients had stage I or II pancreatic cancer. All studies used the internationally accepted TNM staging system. Four studies included only stage I patients, three included only stage II patients, and the remaining 36 studies included both stages. The control groups consisted of 16,577 individuals, including patients with chronic pancreatitis, benign pancreatic diseases, and healthy volunteers. Specifically, 28 studies exclusively used healthy volunteers as controls, whereas 15 studies included both healthy volunteers and patients with chronic pancreatitis or benign pancreatic diseases as controls (Tables 1 and 2).

Regarding biomarker types, seven studies investigated ctDNA [20–26], seven focused on miRNAs [27–33], and seven assessed metabolites [34–39] (Table 3). Most studies (n = 27) evaluated more than one biomarker. Sixteen studies focused on a single biomarker. One metabolite study [36] reported on two independent cohorts from China and the United States, treated separately in our analysis. Additionally, 17 studies evaluated protein biomarkers [40–56]. Five studies [57–61] exclusively examined combining novel biomarkers with CA19–9 and were included in subgroup analyses.

### Quality assessment

Using the QUADAS-2 tool (Fig 2), we found that 20 studies had a high risk of patient selection bias due to potential case-control designs and unclear enrollment processes. Twelve studies had ambiguous descriptions in this domain.

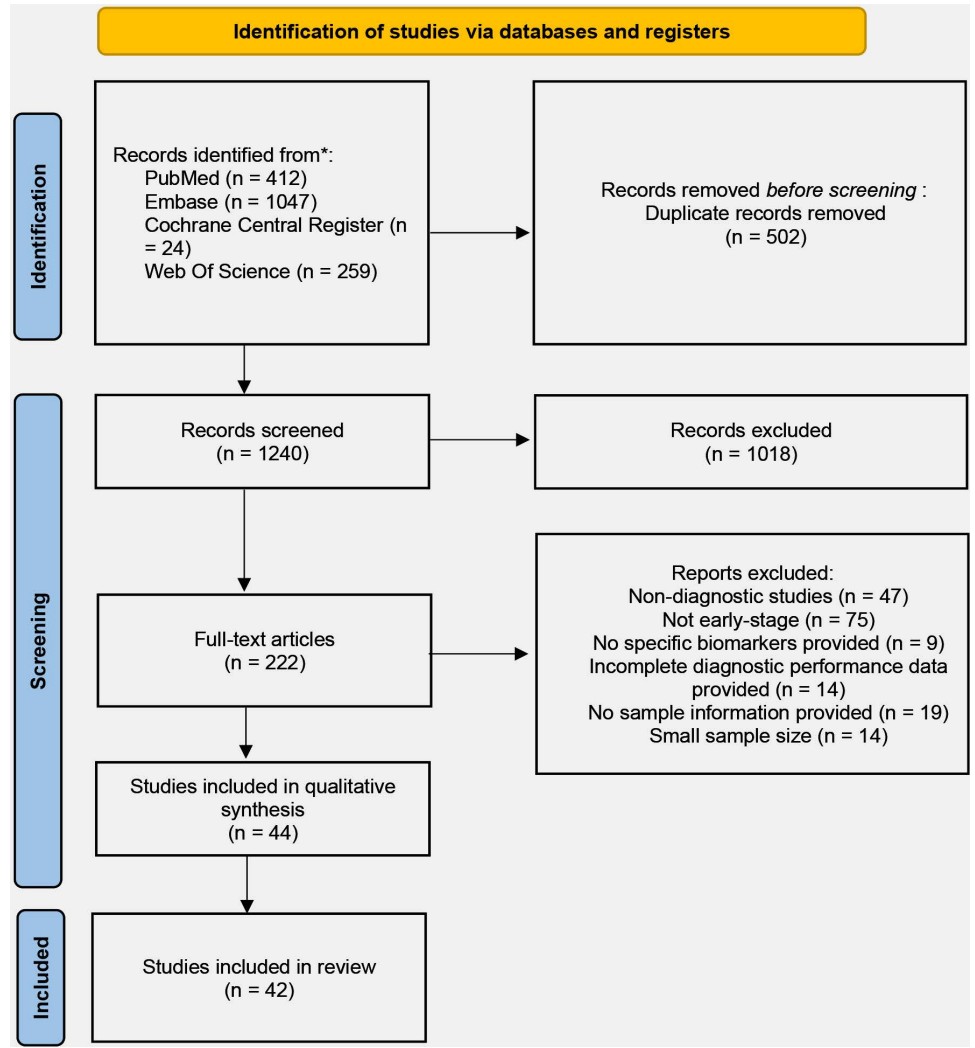

**Fig 1. PRISMA flowchart illustrating the process of screening and selection of studies.**

One study had a high risk of bias in the index test domain due to unclear interpretation processes. All studies had a well-defined reference standard, resulting in a low risk of bias for that domain. All studies were rated low risk in the flow and timing domain due to appropriate intervals between index tests and reference standards.

## Diagnostic accuracy of protein biomarkers

We included 17 studies assessing protein biomarkers for early pancreatic cancer detection (Fig 3). Using a bivariate random-effects model, the pooled sensitivity was 0.79 (95% CI, 0.70–0.86), and the pooled specificity was 0.88 (95% CI, 0.82–0.93). Forest plots showed substantial variability, with I² values of 83.61% for sensitivity and 92.79% for specificity.

The SROC curve had an AUC of 0.90 (95% CI, 0.87–0.93), demonstrating excellent diagnostic accuracy. The pooled DOR was 27.74 (95% CI, 14.32–53.76). Significant heterogeneity was observed, emphasizing the need to investigate variability sources.

**Table 1. Baseline Demographic and Geographic Characteristics of Study Populations.**

| author | year | age | Gender, men(%) | region |
|---|---|---|---|---|
| **Fujimoto** | 2021 | Healthy 55 (34–88)<br>Benign pancreatic disease 69.5 (54–80)<br>Pancreatic cancer 71 (51–92) | Healthy 54<br>Benign pancreatic disease 83<br>Pancreatic cancer 53 | Japan |
| **Majumder** | 2021 | Pancreatic cancer 72.3 (66.0-78.0)<br>Healthy 69.9 (60.8-75.4) | Pancreatic cancer 66<br>Healthy 66 | USA |
| **Wu** | 2022 | Pancreatic cancer 63.1 (35–80)<br>Healthy 51.1 (18–79) | Pancreatic cancer 60.9<br>Healthy 40.5 | China |
| **Bauden** | 2015 | Pancreatic cancer 69 (46–78)<br>Healthy 58 (48–70)<br>Benign disease 72 (58–77) | NA | Sweden |
| **Henriksen** | 2016 | Pancreatic cancer 66 (45–85)<br>Control groups 58 (22–87) | Pancreatic cancer 60<br>Control groups 63.71 | Denmark |
| **Eissa** | 2019 | Pancreatic cancer 60.1 (29–83)<br>Control 65.5 (21–96)<br>Pancreatitis 46.6 (29–70) | Pancreatic cancer 67<br>Control 43<br>Pancreatitis 50 | USA |
| **Cohen** | 2017 | Pancreatic cancer 66.9±10.5 | Pancreatic cancer 54.8 | USA |
| **Debernardi** | 2015 | Healthy 60.7<br>Pancreatic cancer 64.1 | Healthy 46.2<br>Pancreatic cancer 54.3 | UK |
| **Masterson** | 2023 | Pancreatic cancer 65 (18–90)<br>Control 56 (39–80) | Pancreatic cancer 57.3<br>Control 48.3 | USA |
| **Yu** | 2020 | Pancreatic cancer 62.26±9.51<br>Control 59.98±7.14 | Pancreatic cancer 36.6<br>Control 50 | China |
| **Nakamura** | 2022 | Pancreatic cancer 67.6 (±7.9)<br>Control 41.0 (±9.1) | Pancreatic cancer 15 (45.5%)<br>Control 41 (66.1%) | USA/Japan/Korea |
| **Huang** | 2024 | Pancreatic cancer 60<br>Healthy 45<br>Pancreatitis 55<br>pancreatic cystic neoplasms 53 | Pancreatic cancer 56.6<br>Healthy 54.5<br>Pancreatitis 62.2<br>pancreatic cystic neoplasms 32.1 | China |
| **Schultz** | 2014 | Pancreatic cancer 67<br>Healthy 51<br>Pancreatitis 55 | Pancreatic cancer 58.4<br>Healthy 47.4<br>Pancreatitis 64 | Denmark |
| **Ganepola** | 2014 | Pancreatic cancer 68 (62–79)<br>Healthy 46 (42–49)<br>High risk 48 (46–50) | Pancreatic cancer 54<br>Healthy 54<br>High risk 27 | USA |
| **Wolrab** | 2021 | Healthy 65±4<br>Pancreatic cancer 67±4 | NA | Czech Republic |
| **Fukutake** | 2015 | Pancreatic cancer 66.7±9.1<br>Healthy 52.2±10.0<br>Pancreatitis 61.5±13.7 | Pancreatic cancer 62.9<br>Healthy 63.8<br>Pancreatitis 57.1 | Japan |
| **Xie** | 2015 | Pancreatic cancer 67.9 (44.2-85.7)<br>Control 67.8 (44.1-84.3) | Pancreatic cancer 49<br>Control 49 | USA |
| **Xie** | 2015 | Pancreatic cancer 64.3 (40.5-79.1)<br>Control 64.4 (41.2-79.2) | Pancreatic cancer 50<br>Control 50 | China |
| **Zhang** | 2014 | Pancreatic cancer 57.1 (35–78)<br>Healthy 57.8 (34–81)<br>Pancreatitis 55.6 (35–79) | Pancreatic cancer 51.6<br>Healthy 55.1<br>Pancreatitis 52.5 | China |
| **Mayerle** | 2017 | Healthy 69(61–74)<br>Pancreatic cancer 68(55–74) | Healthy 52.5<br>Pancreatic cancer 46.8 | Germany |
| **Hirata** | 2017 | Healthy 65(45–86)<br>Pancreatic cancer 69(63–85) | Healthy 46.6<br>Pancreatic cancer 63.0 | Japan |
| **Yamada** | 2019 | Healthy 56.2 (40–75)<br>Pancreatic cancer 69.7 (25–96) | Healthy 46.5<br>Pancreatic cancer 38.2 | Japan |

*(Continued)*

  

**Table 1.** (Continued)

| author | year | age | Gender, men(%) | region |
|--------|------|-----|----------------|--------|
| **Wang** | 2014 | Healthy 102(≤45) 153(46–55) 121(56–65) 124(>65) Pancreatic cancer 111(≤45) 215(46–55) 272(56–65) 209(>65) | Healthy 57.4 Pancreatic cancer 54.3 | China |
| **Matsunaga** | 2017 | Healthy 60(23–81) Pancreatic cancer 70(39–91) | Healthy 36.1 Pancreatic cancer 63.8 | USA/Japan |
| **Zhou** | 2018 | Pancreatic cancer 83(>60) 73(≤60) Healthy 87(>60) 76(≤60) Pancreatitis 3(>60) 13(≤60) benign pancreatic tumor 4(>60) 16(≤60) | Pancreatic cancer 59.6 Healthy 58.3 Pancreatitis 100 benign pancreatic tumor 20 | China |
| **Sato** | 2020 | Pancreatic cancer 69.0 (21–86) Control 52.0 (22–89) | Pancreatic cancer 69.5 Control 52.1 | Japan |
| **Mawaribuchi** | 2023 | Healthy 50.7(22–70) Pancreatic cancer 67.9(36–92) | Healthy 76.8 Pancreatic cancer 59.1 | Japan |
| **Nam** | 2022 | Healthy 36.0 (18–73) Early-stage Pancreatic cancer 65.0 (44–86) | Healthy 54.1 Early-stage Pancreatic cancer 65.8 | Korea |
| **Han** | 2015 | Pancreatic cancer 61.7±10.8 Healthy 59.1±11.0 Pancreatitis 57.6±11.3 benign pancreatic tumor 58.8±12.1 | Pancreatic cancer 62.9 Healthy 62.5 Pancreatitis 84.6 benign pancreatic tumor 66.7 | China |
| **Kashiro** | 2024 | Healthy 63.2 (50–89) Pancreatic cancer 65.2 (38–86) | Healthy 50.9 Pancreatic cancer 56.6 | Japan |
| **Yu** | 2021 | Healthy 70.7±12.3 Pancreatic cancer 61.9±12.9 | Healthy 55.6 Pancreatic cancer 59.5 | Sweden/Spain |
| **Mohamed** | 2015 | Healthy 62 (46–81) Pancreatic cancer 63 (47–80) | Healthy 55 Pancreatic cancer 64 | Egypt |
| **Radon** | 2015 | Healthy 55 (28–87) Pancreatic cancer 68 (29–94) | Healthy 52.9 Pancreatic cancer 59.4 | UK/Spain/USA |
| **Li** | 2023 | Healthy 61.6 Pancreatic cancer 61.7 | Healthy 58.8 Pancreatic cancer 58.8 | China |
| **Lee. M** | 2021 | Pancreatic cancer 60.4±10.5 Healthy 53.7±14.5 Pancreatitis 48.3±14.0 | Pancreatic cancer 57.1 Healthy 72 Pancreatitis 80 | Korea |
| **Brand** | 2022 | Median age (yr) Healthy 49 Pancreatic cancer 70 | Healthy 51 Pancreatic cancer 58 | USA/Sweden/ Spain/Belgium/ Finland |
| **Aronsson** | 2018 | Healthy 60 (56–63) Pancreatic cancer 68 (64–71) | Healthy 68 Pancreatic cancer 49 | Sweden |
| **Wen** | 2024 | Healthy 57 (43, 71) Pancreatic cancer 64 (54.74) | Healthy 43.2 Pancreatic cancer 63.4 | China |
| **Haab** | 2024 | Pancreatic cancer 64.3 (40.5-79.1) Control 64.4 (41.2-79.2) | Pancreatic cancer 50.6 Control 41.6 | USA |
| **Kim** | 2020 | Healthy 57.5 (8.6) Pancreatic cancer 66.0 (9.5) | Healthy 60.7 Pancreatic cancer 70 | Korea |
| **Lee. D** | 2021 | Healthy 56.4±11.0 Pancreatic cancer 63.1±9.9 | Healthy 52.9 Pancreatic cancer 60.8 | Korea |
| **Firpo** | 2023 | Pancreatic cancer 69 (61–74) Healthy 57 (51–67) Pancreatitis 50 (39–62) intraductal papillary mucinous neoplasm 67 (57–73) | Pancreatic cancer 47 Healthy 50 Pancreatitis 53 intraductal papillary mucinous neoplasm 39 | USA/Italy |
| **Zhang** | 2013 | Healthy 53.74 (19–85) Pancreatic cancer 54.36 (18–85) | Healthy 51.9 Pancreatic cancer 60.5 | China |

**Table 2. Study Characteristics and Diagnostic Outcomes.**

| author | year | tp | fp | fn | tn | sample size | stage | control |
|---|---|---|---|---|---|---|---|---|
| Fujimoto | 2021 | 5 | 6 | 4 | 86 | 101 | StageI | NC[a] |
| Majumder | 2021 | 34 | 3 | 16 | 47 | 100 | StageI and II | HC[b] |
| Wu | 2022 | 49 | 4 | 20 | 33 | 106 | StageI and II | HC |
| Bauden | 2015 | 10 | 3 | 15 | 31 | 59 | StageII | NC |
| Henriksen | 2016 | 29 | 21 | 11 | 103 | 164 | StageI and II | HC |
| Eissa | 2019 | 36 | 8 | 1 | 87 | 132 | StageI and II | HC |
| Cohen | 2017 | 66 | 1 | 155 | 181 | 403 | StageI and II | HC |
| Debernardi | 2015 | 5 | 1 | 1 | 25 | 32 | StageI | HC |
| Masterson | 2023 | 34 | 0 | 0 | 60 | 94 | StageI and II | HC |
| Yu | 2020 | 33 | 6 | 9 | 85 | 133 | StageI and II | NC |
| Nakamura | 2022 | 40 | 3 | 10 | 30 | 83 | StageI and II | HC |
| Huang | 2024 | 98 | 81 | 5 | 271 | 455 | StageI and II | NC |
| Schultz | 2014 | 56 | 68 | 14 | 279 | 417 | StageI and II | NC |
| Ganepola | 2014 | 10 | 2 | 1 | 20 | 33 | StageII | NC |
| Wolrab | 2021 | 38 | 1 | 1 | 38 | 78 | StageI and II | HC |
| Fukutake | 2015 | 54 | 1554 | 30 | 6218 | 7856 | StageII | HC |
| Xie | 2015 | 56 | 17 | 10 | 83 | 166 | StageI and II | NC |
| Xie | 2015 | 77 | 24 | 23 | 76 | 200 | StageI and II | NC |
| Zhang | 2014 | 13 | 27 | 2 | 239 | 281 | StageI and II | NC |
| Mayerle | 2017 | 36 | 15 | 4 | 65 | 120 | StageI and II | HC |
| Hirata | 2017 | 38 | 6 | 16 | 52 | 112 | StageI and II | HC |
| Yamada | 2019 | 9 | 8 | 6 | 35 | 58 | StageI | HC |
| Wang | 2014 | 112 | 18 | 60 | 482 | 672 | StageI and II | HC |
| Matsunaga | 2017 | 22 | 14 | 4 | 47 | 87 | StageI and IIA | HC |
| Zhou | 2018 | 67 | 68 | 5 | 131 | 271 | StageI and II | NC |
| Sato | 2020 | 11 | 395 | 6 | 3766 | 4178 | StageI and II | NC |
| Mawaribuchi | 2023 | 26 | 18 | 3 | 81 | 128 | StageI and IIA | HC |
| Nam | 2022 | 106 | 13 | 5 | 186 | 310 | StageI and II | HC |
| Han | 2015 | 53 | 19 | 9 | 73 | 154 | StageI and II | NC |
| Kashiro | 2024 | 15 | 5 | 15 | 101 | 136 | StageI and II | HC |
| Yu | 2021 | 20 | 1 | 17 | 35 | 73 | StageI and II | HC |
| Mohamed | 2015 | 19 | 11 | 2 | 9 | 41 | StageI and II | HC |
| Radon | 2015 | 12 | 6 | 3 | 20 | 41 | StageI and II | HC |
| Li | 2023 | 36 | 25 | 31 | 145 | 237 | StageI and II | HC |
| Lee. M | 2021 | 33 | 3 | 2 | 62 | 100 | StageI and II | NC |
| Brand | 2022 | 49 | 2 | 7 | 214 | 272 | StageI and II | HC |
| Aronsson | 2018 | 7 | 9 | 2 | 38 | 56 | StageI | HC |
| Wen | 2024 | 43 | 18 | 8 | 70 | 139 | StageI and II | HC |
| Haab | 2024 | 122 | 7 | 48 | 118 | 295 | StageI and II | NC |
| Kim | 2020 | 38 | 8 | 4 | 183 | 233 | StageI and II | HC |
| Lee. D | 2021 | 180 | 16 | 12 | 167 | 375 | StageI and II | HC |
| Firpo | 2023 | 26 | 3 | 6 | 69 | 104 | StageI and II | NC |
| Zhang | 2013 | 54 | 18 | 4 | 165 | 241 | StageI and II | HC |

a NC: Non-cancer Control group (including pancreatitis, benign pancreatic diseases).

b HC: Healthy Control group.

**Table 3. Biomarker Types, Detection Methods, and Sample Types.**

| author | year | biomaker type | biomarker name | detection methods | speci-mens |
|---|---|---|---|---|---|
| **Fujimoto** | 2021 | ctDNA | Methylated RUNX3 | CORD assay (Combined Restriction Digital PCR) | blood |
| **Majumder** | 2021 | ctDNA | Thirteen methylated DNA markers (GRIN2D, CD1D, ZNF781, FER1L4, RYR2, CLEC11A, AK055957, LRRC4, GH05J042948, HOXA1, PRKCB, SHISA9, and NTRK3) | plasma TELQAS (target enrichment with long probe quantitative amplified signal) assays | plasma |
| **Wu** | 2022 | ctDNA | methylation signature of circulating tumour DNA | Reduced representation bisulfite sequencing(RRBS) and targeted methylation sequencing | plasma |
| **Bauden** | 2015 | ctDNA | 5-methylcytosine | Nucleosomics, a novel ELISA method. | serum |
| **Henriksen** | 2016 | ctDNA | Cell-free DNA promoter hypermethylation in plasma(BMP3, RASSF1A, BNC1, MESTv2, TFPI2, APC, SFRP1 and SFRP2) | methylation-specific PCR | plasma |
| **Eissa** | 2019 | ctDNA | promoter DNA methylation of the genes ADAMTS1 and BNC1 | methylation on beads (MOB) | blood |
| **Cohen** | 2017 | ctDNA | ctDNA | Safe-SeqS | plasma |
| **Debernardi** | 2015 | miRNA | miRNA biomarkers(miR-143, miR-223, miR-204, miR-30e) | RT-PCR | urinary |
| **Masterson** | 2023 | miRNA | microRNA-10b+microRNA-let7a | nanoplasmonic sensors | plasma |
| **Yu** | 2020 | miRNA | miRNA-25 | RT-PCR | serum |
| **Nakamura** | 2022 | miRNA | 5 cf-miRNAs (miR30c-5p, miR340-5p, miR335-5p, miR23b-3p and miR142-3p) and 8 exo-miRNA candidates (miR145-5p, miR200b-3p, miR429, miR1260b, miR145-3p, miR216b-5p, miR200a-3p and miR217-5p) | qRT-PCR | blood |
| **Huang** | 2024 | miRNA | miRNA Panel(hsa-miR-132-3p, hsa-miR-30c-5p, hsa-miR-24-3p, hsa-miR-23a-3p) | RT-PCR | Serum |
| **Schultz** | 2014 | miRNA | miR-26b, miR-34a, miR-122, miR-126*, miR-145, miR-150, miR-223, miR-505, miR-636, miR-885.5p | qPCR and TaqMan assays | blood |
| **Ganepola** | 2014 | miRNA | miR-642b-3p, miR-885-5p, miR-22-3p | RT-qPCR | blood |
| **Wolrab** | 2021 | Metabolite | serum lipids(sphingomyelins (SM), ceramides (Cer), phosphatidylcholines (PC), lysophosphatidylcholines (LPC)) | UHPSFC/MS, shotgun MS and MALDI-MS | serum |
| **Fukutake** | 2015 | Metabolite | six plasma free amino acids: serine, asparagine, isoleucine, alanine, histidine, and tryptophan | HPLC-MS | plasma |
| **Xie** | 2015 | Metabolite | glutamate, choline, 1,5-anhydro-D-glucitol, betaine, and methylguanidine | LC-TOFMS and GC-TOFMS | plasma |
| **Xie** | 2015 | Metabolite | glutamate, choline, 1,5-anhydro-D-glucitol, betaine, and methylguanidine | LC-TOFMS and GC-TOFMS | plasma |
| **Zhang** | 2014 | Metabolite | serum unsaturated fatty acids(Panel a: C16:1 Palmitoleic acid, C18:3 linolenic acid, C18:2 linoleic acid, C20:4 arachidonic acid and C22:6 docosahexaenoic acid) | CBDInanoESI-FTICR MS | serum |
| **Mayerle** | 2017 | Metabolite | nine plasma metabolites(Proline, Sphingomyelin (d18:2,C17:0), Phosphatidylcholine (C18:0,C22:6), Isocitrate, Sphinganine-1-phosphate (d18:0), Histidine, Pyruvate, Ceramide (d18:1,C24:0), Sphingomyelin (d17:1,C18:0)) | GC-MS, LC-MS/MS | blood |
| **Hirata** | 2017 | Metabolite | metabolites(histidine, xylitol) | GC/MS/MS | serum |
| **Yamada** | 2019 | Protein | anti-3'-sialyllactose antibodies | ELISA | serum |
| **Wang** | 2014 | Protein | Macrophage inhibitory cytokine 1 | RT-PCR and ELISA | serum |
| **Matsunaga** | 2017 | Protein | S100P in Duodenal Fluid | ELISA | Duodenal fluid |
| **Zhou** | 2018 | Protein | serum glypican-1 | ELISA | serum |
| **Sato** | 2020 | Protein | Apolipoprotein A2 isoforms (apoA2-i) | ELISA | plasma |

*(Continued)*

**Table 3.** (Continued)

| author | year | biomaker type | biomarker name | detection methods | speci-mens |
|---|---|---|---|---|---|
| **Mawaribuchi** | 2023 | Protein | rBC2LCN-reactive SERPINA3 | Lectin-ELISA, Liquid Chromatography-Tandem Mass Spectrometry (LC-MS/MS) | serum |
| **Nam** | 2022 | Protein | Asprosin | ELISA | serum |
| **Han** | 2015 | Protein | Serum dickkopf-1 | ELISA | serum |
| **Kashiro** | 2024 | Protein | apolipoprotein-A2 isoforms | ELISA | blood |
| **Yu** | 2021 | Protein | selected protein panels: S100A11, PPY, RET, 5'-NT, ITGB5, ERBB3, SCAMP3 and CEACAM1 | Proximity Extension Assay(PEA),ELISA, ECLIA | plasma |
| **Mohamed** | 2015 | Protein | MIC-1 | ELISA | serum |
| **Radon** | 2015 | Protein | LYVE-1, REG1A, and TFF1 in Urine | GeLC/MS/MS analysis and ELISA assays | Urine |
| **Li** | 2023 | Protein | A panel of three TAAbs (anti-HEXB, anti-TXLNA, and anti-SLAMF6) | ELISA | serum |
| **Lee. M** | 2021 | Protein | Complement Factor B (CFB) | ELISA, ECLIA | plasma |
| **Brand** | 2022 | Protein | Single-chain variable fragment antibody: A1026, A1048, A1065, PC105, PC150, PC157, PC165, PC242 | IMMray PanCan-d test | blood |
| **Aronsson** | 2018 | Protein | IL-17E, B7-1, DR6 | RayBio Human Glycosyla-tion Antibody Array 1000, RayBiotech | serum |
| **Wen** | 2024 | Protein | N-glycan biosignatures | Serum glycoprotein N-glycome profiling | serum |
| **Haab** | 2024 | Protein | CA199STRA+MUC16STRA+CA19–9 | Antibody microarrays, Luminex bead-based immunoassays and sand-wich ELISA assay | plasma |
| **Kim** | 2020 | Protein | ApoA1、CA125、CA19–9、CEA、ApoA2 and TTR | immunoturbidimetric method, electrochemilumi-nescent detection method, ELISA | blood |
| **Lee. D** | 2021 | Protein | LRG1, TTR, CA 19–9 | ELISA | blood |
| **Firpo** | 2023 | Protein | 31 analytes Serum Biomarker Panel: ALCAM, ANG, AXL, BAG3, BSG, CA 19–9, CEA, CEACAM1, COL18A1, EPCAM, HA, HP, ICAM1, IGFBP2, IGFBP4, LCN2, LRG1, MMP2, MMP7, MMP9, MSLN, PARK7, PPBP, PRG4, SPARCL1, SPP1, TGFBI, THBS1, TIMP1, TNFRSF1A, VEGFC | ELISA | serum |
| **Zhang** | 2013 | Protein | CA19–9, ALB, CRP and IL-8 | ELISA | serum |

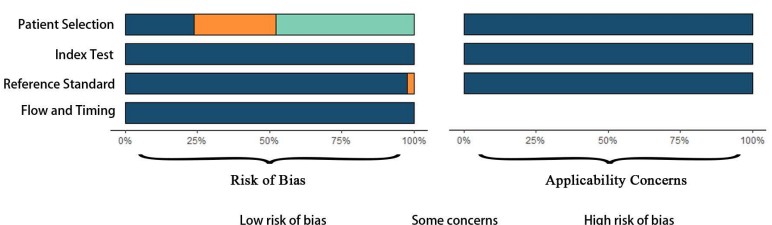

**Fig 2. Risk of Bias and Applicability Concerns Assessment Using QUADAS-2.**

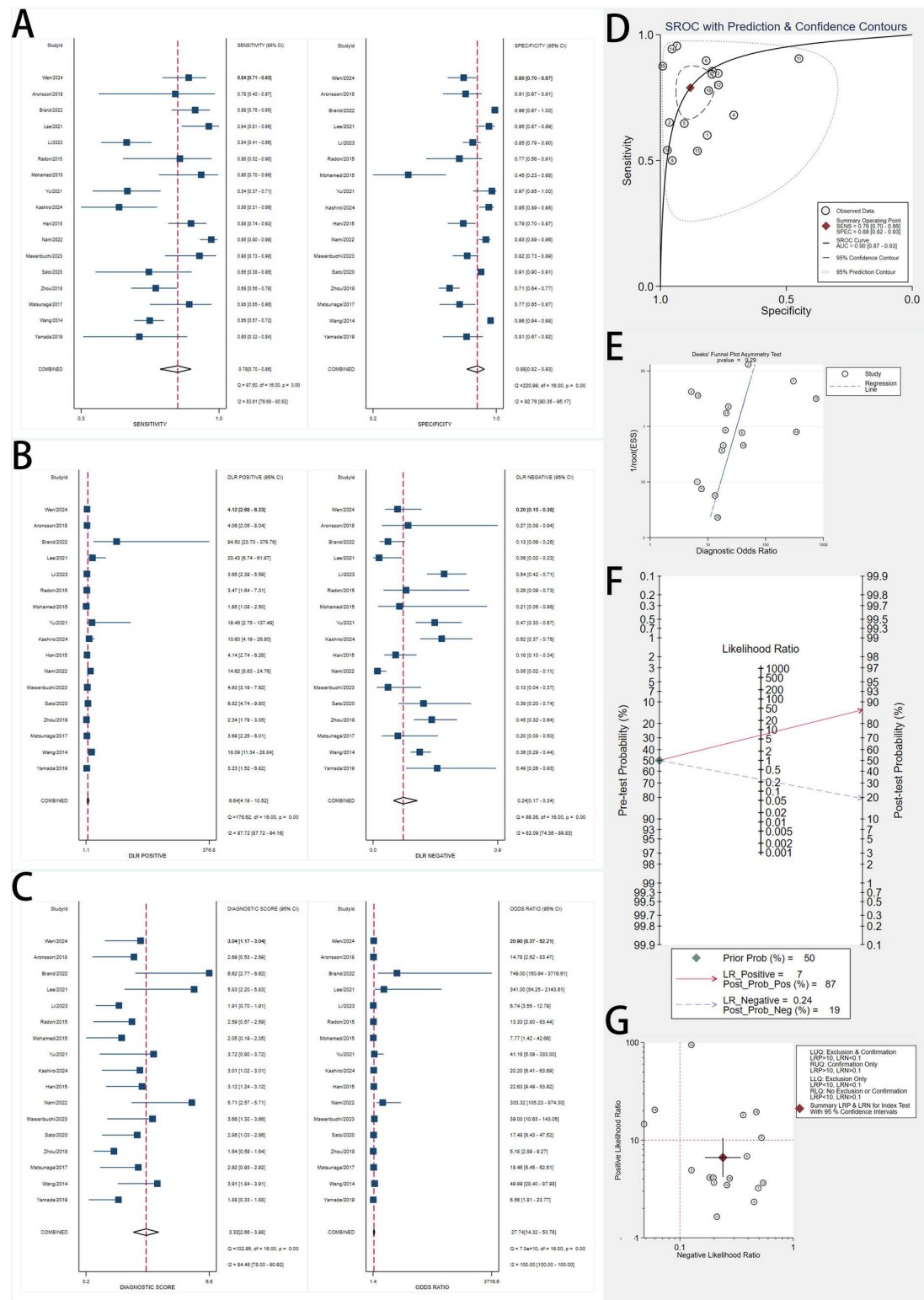

**Fig 3. Forest Plots, SROC Curve, and Diagnostic Performance Analyses of Protein Biomarkers for Early Pancreatic Cancer Detection.** A: Forest plots showing pooled sensitivity (left) and specificity (right) estimates of included studies with 95% confidence intervals (CIs). The red dashed lines indicate the overall pooled sensitivity and specificity. B: Forest plots of positive likelihood ratios (LR+) and negative likelihood ratios (LR-) with

pooled estimates. LR+ measures the likelihood of a positive test result in patients with the disease, while LR- assesses the likelihood of a negative test in patients without the disease. C: Forest plots of diagnostic score (log diagnostic odds ratio) and diagnostic odds ratio (DOR) with corresponding 95% CIs. Higher DOR values indicate stronger overall diagnostic accuracy. D: Summary Receiver Operating Characteristic (SROC) curve with prediction and confidence contours, depicting the diagnostic accuracy of biomarkers across studies. The red point represents the summary operating point, and the dotted lines indicate the confidence and prediction intervals. E: Deeks' funnel plot asymmetry test for publication bias assessment. A non-significant p-value (p>0.05) suggests no evidence of significant publication bias. F: Fagan's nomogram demonstrating post-test probabilities based on a pretest probability of 50%. Positive tests increase post-test probability to 87%, while negative tests reduce it to 19%. G: Scatter plot of positive likelihood ratio (PLR) versus negative likelihood ratio (NLR) across studies, illustrating variability in diagnostic accuracy.

The pooled PLR was 6.64 (95% CI, 4.19–10.52), and the pooled NLR was 0.24 (95% CI, 0.17–0.34). Using a pretest probability of 50%, Fagan's nomogram showed that a positive test increased the posttest probability to 87%, while a negative test reduced it to 19%.

Deeks' funnel plot asymmetry test yielded a p-value of 0.29, indicating no significant publication bias.

### Diagnostic performance of ctDNA biomarkers

Seven studies evaluated ctDNA biomarkers (Fig 4). The pooled sensitivity was 0.65 (95% CI, 0.48–0.81), and the pooled specificity was 0.94 (95% CI, 0.88–0.97). Forest plots revealed significant heterogeneity, with I² values of 94.33% for sensitivity and 85.08% for specificity.

The SROC curve showed an AUC of 0.92 (95% CI, 0.89–0.94). The pooled DOR was 27.73 (95% CI, 12.91–59.55). The pooled PLR was 10.26 (95% CI, 5.75–18.25), and the pooled NLR was 0.37 (95% CI, 0.22–0.62). Fagan's nomogram indicated that a positive ctDNA result increased the posttest probability to 91%, while a negative result decreased it to 27%.

Deeks' funnel plot asymmetry test showed no significant publication bias (p=0.29).

### Diagnostic performance of miRNA biomarkers

Seven studies assessed miRNA biomarkers (Fig 5). The pooled sensitivity was 0.88 (95% CI, 0.79–0.93), and the pooled specificity was 0.91 (95% CI, 0.82–0.95). High heterogeneity was observed, with I² values of 77.36% for sensitivity and 91.27% for specificity.

The SROC curve had an AUC of 0.95 (95% CI, 0.93–0.97). The pooled DOR was 72.68 (95% CI, 26.64–198.24). The pooled PLR was 9.57 (95% CI, 4.85–18.88), and the pooled NLR was 0.13 (95% CI, 0.07–0.24). Fagan's nomogram showed that a positive miRNA test increased the posttest probability to 91%, while a negative result reduced it to 12%.

Deeks' funnel plot asymmetry test indicated no significant publication bias (p=0.45).

### Diagnostic accuracy of metabolite biomarkers

Seven studies evaluated metabolite biomarkers (Fig 6). The pooled sensitivity was 0.84 (95% CI, 0.73–0.92), and the pooled specificity was 0.85 (95% CI, 0.81–0.88). Significant heterogeneity was present, with I² values of 81.17% for sensitivity and 98.05% for specificity.

The SROC curve showed an AUC of 0.90 (95% CI, 0.87–0.93). The pooled DOR was 31.76 (95% CI, 12.38–81.48). The pooled PLR was 5.80 (95% CI, 4.10–8.20), and the pooled NLR was 0.18 (95% CI, 0.10–0.34). Fagan's nomogram indicated that a positive test increased the posttest probability to 85%, while a negative result reduced it to 15%.

Deeks' funnel plot asymmetry test suggested potential publication bias (p=0.04), so these results should be interpreted with caution.

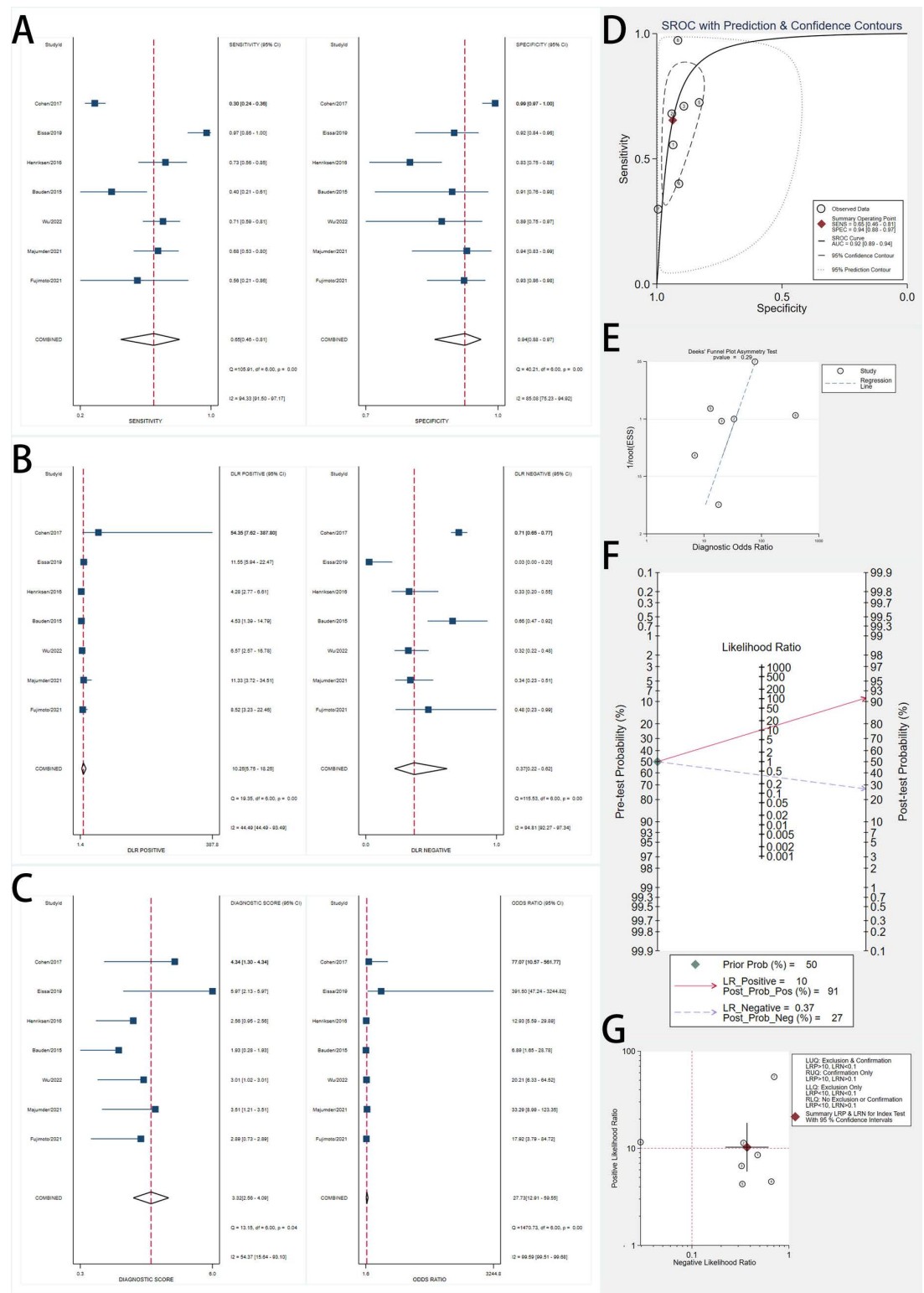

**Fig 4. Forest Plots, SROC Curve, and Diagnostic Performance Analyses of ctDNA Biomarkers for Early Pancreatic Cancer Detection.** A: Forest plots showing pooled sensitivity (left) and specificity (right) estimates of included studies with 95% confidence intervals (CIs). The red dashed lines indicate the overall pooled sensitivity and specificity. B: Forest plots of positive likelihood ratios (LR+) and negative likelihood ratios (LR-) with pooled estimates. LR+ measures the likelihood of a positive test result in patients with the disease, while LR- assesses the likelihood of a negative test in patients

without the disease. C: Forest plots of diagnostic score (log diagnostic odds ratio) and diagnostic odds ratio (DOR) with corresponding 95% CIs. Higher DOR values indicate stronger overall diagnostic accuracy. D: Summary Receiver Operating Characteristic (SROC) curve with prediction and confidence contours, depicting the diagnostic accuracy of biomarkers across studies. The red point represents the summary operating point, and the dotted lines indicate the confidence and prediction intervals. E: Deeks' funnel plot asymmetry test for publication bias assessment. A non-significant p-value (p > 0.05) suggests no evidence of significant publication bias. F: Fagan's nomogram showing post-test probabilities for a pretest probability of 50%. A positive test increases the post-test probability to 91%, while a negative test decreases it to 27%. G: Scatter plot of positive likelihood ratio (PLR) versus negative likelihood ratio (NLR) across studies, illustrating variability in diagnostic accuracy.

## Subgroup analysis results

We conducted subgroup analyses on proteins, ctDNA, miRNAs, and metabolites. When novel biomarkers are combined with CA19–9, the diagnostic performance exceeds what can be achieved by using these biomarkers alone. In this study, we began by conducting a meta-analysis to determine the overall performance of novel biomarkers used in conjunction with CA19–9. We then closely examined the definitions and calculation methods employed in the relevant articles and observed two primary strategies.

The first strategy, known as the "either-positive" approach, calculates sensitivity by classifying a patient's overall result as positive if either CA19–9 or the novel biomarker is positive. Specificity, on the other hand, is determined by requiring both tests to be negative—referred to as "joint negativity." In other words, this approach applies an "OR" principle for positivity and an "AND" principle for negativity. By contrast, some studies utilized regression models, scoring systems, or machine learning algorithms (e.g., logistic regression or multi-marker panels) to estimate the combined sensitivity and specificity of a biomarker with CA19–9. Under these methods, an integrated score or a predicted probability serves as a threshold: results above this threshold are classified as positive, and those below are classified as negative. Consequently, in such cases, the reported sensitivity and specificity arise from modeling or receiver operating characteristic (ROC) analysis rather than a simple binary combination. Based on these distinctions, we conducted subgroup analyses for each approach.

Fourteen studies [40–44,47,52,54,56–61] addressed the combination of protein biomarkers with CA19–9 for early pancreatic cancer detection. The pooled analysis showed a sensitivity of 0.90 and a specificity of 0.92. After categorizing the articles by methodology, five [42,43,56,57,62] studies employing the either-positive principle yielded a combined sensitivity of 0.86 and specificity of 0.88 (S1 Fig), whereas nine [40,41,44,47,54,58–61] studies using predictive models or machine learning reported a sensitivity of 0.92 and a specificity of 0.94 (S2 Fig). There was no substantial difference in I² between the two groups. Nevertheless, these results suggest that algorithmic approaches may modestly increase both sensitivity and specificity compared to the simpler either-positive principle.

Another five studies [20,21,23,26,63] investigated ctDNA in combination with CA19–9. Of these, three employed predictive modeling or machine learning, and two adopted the either-positive approach. Because of the limited number of studies, no subgroup meta-analysis was conducted for ctDNA. The pooled results showed a sensitivity of 0.79 and a specificity of 0.95. Although this combination demonstrates relatively high specificity, there is still a need to enhance its sensitivity (Fig 7).

Studies using healthy volunteers as controls showed similar diagnostic performance to those including patients with precancerous conditions or chronic pancreatitis, indicating that these biomarkers may not have the high false-positive rates associated with CA19–9 in certain populations (Fig 8).

Additional analyses specifically examined studies employing ELISA for protein biomarkers and PCR for miRNA biomarkers. Neither the diagnostic accuracy nor the observed heterogeneity demonstrated significant changes within these subgroups (Fig 9).

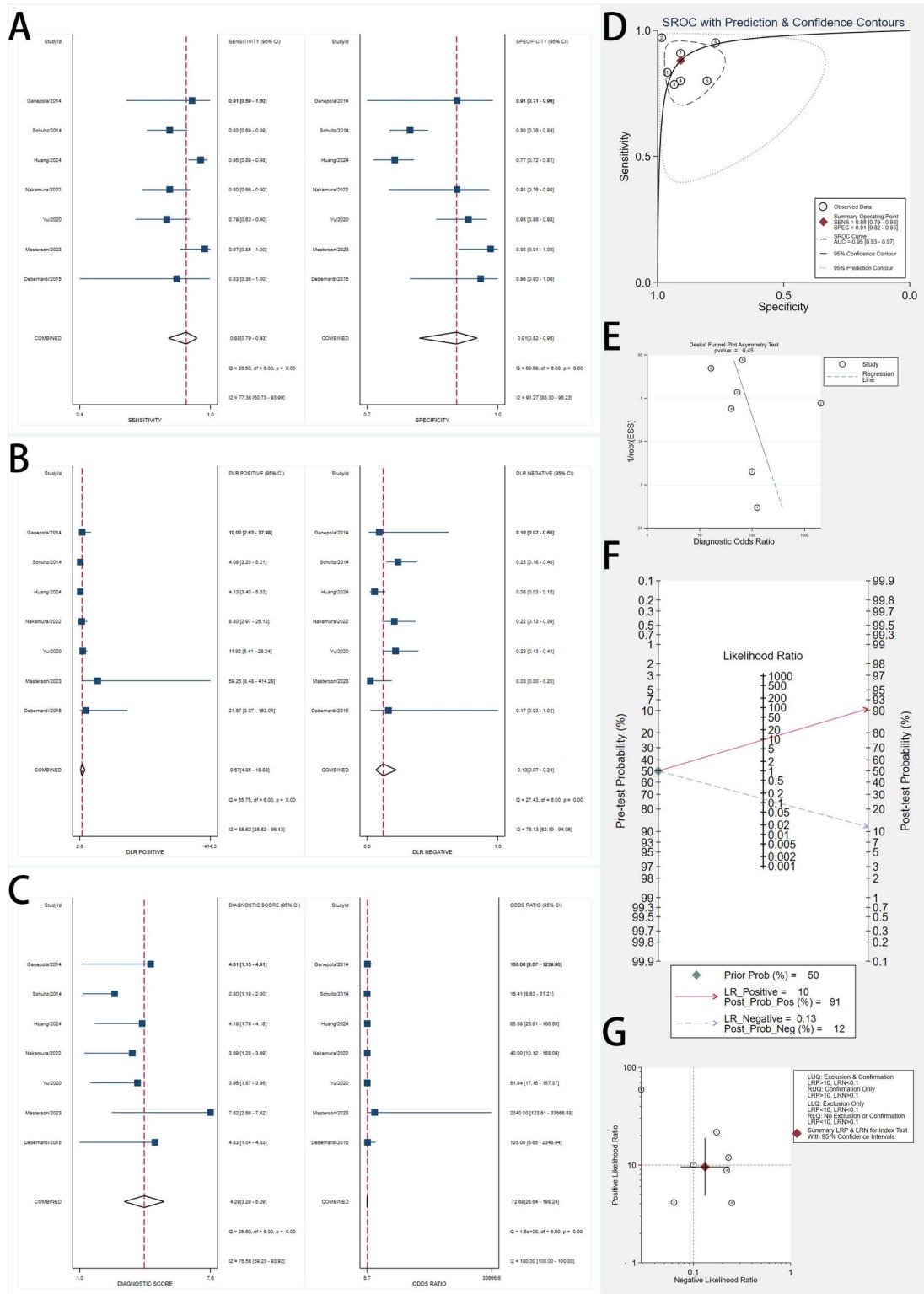

**Fig 5. Forest Plots, SROC Curve, and Diagnostic Performance Analyses of miRNA Biomarkers for Early Pancreatic Cancer Detection.** A: Forest plots showing pooled sensitivity (left) and specificity (right) estimates of included studies with 95% confidence intervals (CIs). The red dashed lines indicate the overall pooled sensitivity and specificity. B: Forest plots of positive likelihood ratios (LR+) and negative likelihood ratios (LR-) with

pooled estimates. LR+ measures the likelihood of a positive test result in patients with the disease, while LR- assesses the likelihood of a negative test in patients without the disease. C: Forest plots of diagnostic score (log diagnostic odds ratio) and diagnostic odds ratio (DOR) with corresponding 95% CIs. Higher DOR values indicate stronger overall diagnostic accuracy. D: Summary Receiver Operating Characteristic (SROC) curve with prediction and confidence contours, depicting the diagnostic accuracy of biomarkers across studies. The red point represents the summary operating point, and the dotted lines indicate the confidence and prediction intervals. E: Deeks' funnel plot asymmetry test for publication bias assessment. A non-significant p-value (p > 0.05) suggests no evidence of significant publication bias. F: Fagan's nomogram demonstrating post-test probabilities based on a pretest probability of 50%. A positive test result increases the post-test probability to 91%, while a negative test reduces it to 12%. G: Scatter plot of positive likelihood ratio (PLR) versus negative likelihood ratio (NLR) across studies, illustrating variability in diagnostic accuracy.

## Sensitivity analysis

We performed sensitivity analyses by excluding studies with high risk of bias and small sample sizes (n ≤ 50). Reanalyzing the data showed no significant changes in pooled estimates. A leave-one-out analysis indicated that no single study unduly influenced the overall results, suggesting robustness (S7 Table).

## Discussion

This meta-analysis evaluated the diagnostic accuracy of novel biomarkers—protein biomarkers, ctDNA, miRNAs, and metabolites—for early pancreatic cancer detection. Our findings show that these biomarkers have good diagnostic performance, with miRNA biomarkers exhibiting the highest pooled sensitivity and specificity, followed by protein and metabolite biomarkers. Although ctDNA biomarkers had high specificity, their sensitivity was moderate; however, sensitivity improved after excluding small-sample studies. These results highlight the potential of these biomarkers in enhancing early detection, crucial for improving patient outcomes.

Our results align with studies reporting high diagnostic accuracy of miRNAs in pancreatic cancer detection [64–67]. The high sensitivity and specificity of miRNA biomarkers may be due to their stability in bodily fluids and role in gene regulation related to carcinogenesis [68]. Protein biomarkers also showed strong diagnostic performance, consistent with research emphasizing their potential in cancer diagnosis [69,70]. The moderate sensitivity but high specificity of ctDNA biomarkers reflect challenges in detecting low levels of tumor-derived DNA in early-stage disease [71]. Metabolite biomarkers demonstrated good diagnostic accuracy, supporting findings that metabolic changes indicate tumor presence and progression [72].

Unlike previous analyses focusing on all pancreatic cancer patients, our study specifically analyzed stage I and II patients. These biomarkers showed stronger diagnostic performance in early-stage diagnosis compared to overall pancreatic cancer populations.

The high sensitivity and specificity of miRNA and protein biomarkers suggest they could be effective noninvasive tools for early detection. miRNAs may reflect early molecular changes in carcinogenesis, and protein biomarkers can be detected through established assays like ELISA, offering practical clinical advantages. The high specificity of ctDNA biomarkers indicates potential for confirming diagnoses, though combining them with other biomarkers or imaging may enhance effectiveness. Metabolite biomarkers provide another detection avenue but may require standardized platforms for consistency.

Incorporating these novel biomarkers into clinical practice could significantly improve early detection rates, increasing eligibility for curative surgery. Combining them with the traditional CA19–9 assay could enhance diagnostic accuracy, as our subgroup analyses showed improved performance with combined use. This approach could mitigate individual biomarker limitations and offer a more comprehensive diagnostic tool. In addition, our study performed a regrouped subgroup

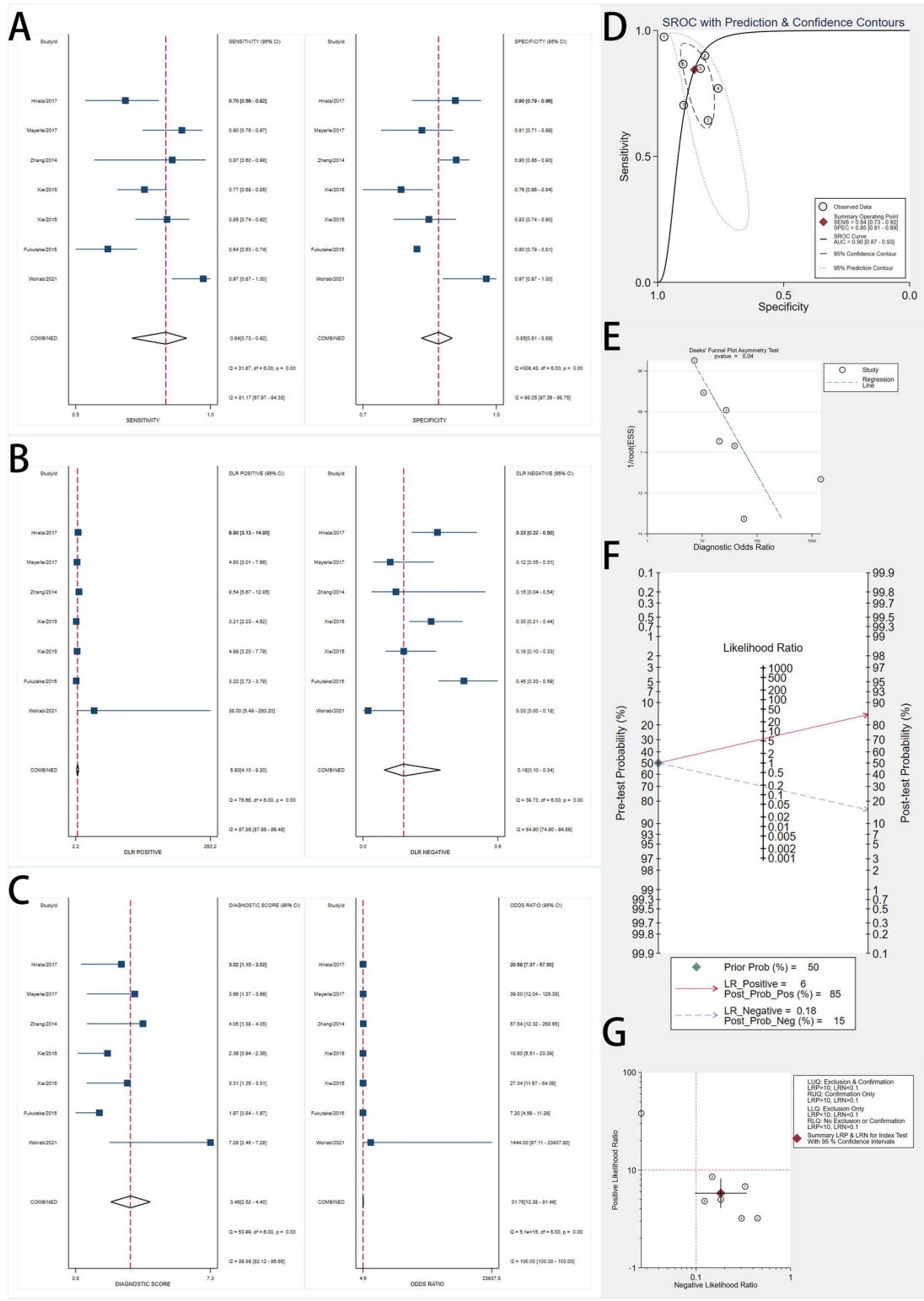

**Fig 6. Forest Plots, SROC Curve, and Diagnostic Performance Analyses of Metabolite Biomarkers for Early Pancreatic Cancer Detection.** A: Forest plots showing pooled sensitivity (left) and specificity (right) estimates of included studies with 95% confidence intervals (CIs). The red dashed lines indicate the overall pooled sensitivity and specificity. B: Forest plots of positive likelihood ratios (LR+) and negative likelihood ratios (LR-) with pooled estimates. LR+ measures the likelihood of a positive test result in patients with the disease, while LR- assesses the likelihood of a negative test

in patients without the disease. C: Forest plots of diagnostic score (log diagnostic odds ratio) and diagnostic odds ratio (DOR) with corresponding 95% CIs. Higher DOR values indicate stronger overall diagnostic accuracy. D: Summary Receiver Operating Characteristic (SROC) curve with prediction and confidence contours, depicting the diagnostic accuracy of biomarkers across studies. The red point represents the summary operating point, and the dotted lines indicate the confidence and prediction intervals. E: Deeks' funnel plot asymmetry test for publication bias assessment. A non-significant p-value ($p > 0.05$) suggests no evidence of significant publication bias. F: Fagan's nomogram demonstrating post-test probabilities. Based on a pretest probability of 50%, a positive test result increases the post-test probability to 85%, while a negative test reduces it to 15%. G: Scatter plot of positive likelihood ratio (PLR) versus negative likelihood ratio (NLR) across studies, illustrating variability in diagnostic accuracy.

analysis of the literature on combining the novel biomarkers with CA19–9. Some studies used an "either-positive" principle—treating a test result as positive if either CA19–9 or the novel biomarker was positive—an approach that tends to increase sensitivity in clinical screening, while applying a "joint negativity principle" to enhance specificity. Other studies employed regression equations, machine learning algorithms, or multi-marker predictive models to integrate multiple variables into a single composite score, using a specified threshold to determine positivity or negativity. The subgroup analysis showed that, for protein biomarkers combined with CA19–9, the overall sensitivity and specificity under the either-positive principle (0.86 and 0.88, respectively) were slightly lower than those achieved by predictive modeling or machine learning methods 0.92 and 0.94, respectively). These findings underscore the importance of algorithmic optimization for improving diagnostic accuracy. They also point to the potential necessity and feasibility of incorporating additional biomarkers or refining detection techniques and algorithms to further enhance diagnostic performance. Our analyses also indicated that diagnostic performance was unaffected by control group composition, suggesting these biomarkers may avoid CA19–9's high false-positive rates in certain populations.

A major strength of this meta-analysis is the comprehensive evaluation across numerous studies, providing a reliable assessment of diagnostic performance. We used strict inclusion criteria and quality assessments, which enhanced the robustness of our findings. The statistical approach we employed was a random-effects meta-analysis model, which accounts for between-study variability by assuming that the true effect sizes may differ across studies. This model was chosen to accommodate the heterogeneity observed in study designs, patient populations, biomarkers, and detection methods. By using this model, we were able to calculate pooled estimates of sensitivity, specificity, positive likelihood ratio (PLR), negative likelihood ratio (NLR), and diagnostic odds ratio (DOR) with 95% confidence intervals. Additionally, we constructed summary receiver operating characteristic (SROC) curves to assess overall diagnostic accuracy.

However, significant heterogeneity existed, likely due to biomarker diversity and differences in study design, patient populations, cancer staging, control groups, and detection methods. Potential publication bias, especially in metabolite studies, suggests caution and may relate to sample size. Sensitivity analyses did not show significant heterogeneity associated with very large or small sample sizes. Despite efforts to explore heterogeneity sources, residual variability remains. Methodological quality varied among studies, which may have influenced the pooled estimates.

Future research should focus on standardizing detection and validation methods through large, multicenter prospective studies. Developing standardized protocols for sample handling is crucial for clinical translation. Although biomarkers were pooled by type to estimate the overall diagnostic accuracy of each class (miRNAs, proteins, ctDNA), we acknowledge that individual biomarkers within each category may exhibit different cut-off values, sensitivities, and specificities. This variability can limit the generalizability of the pooled estimates. Therefore, future studies should prioritize the validation of specific biomarker combinations rather than isolated markers, and standardize detection methods and cut-off values

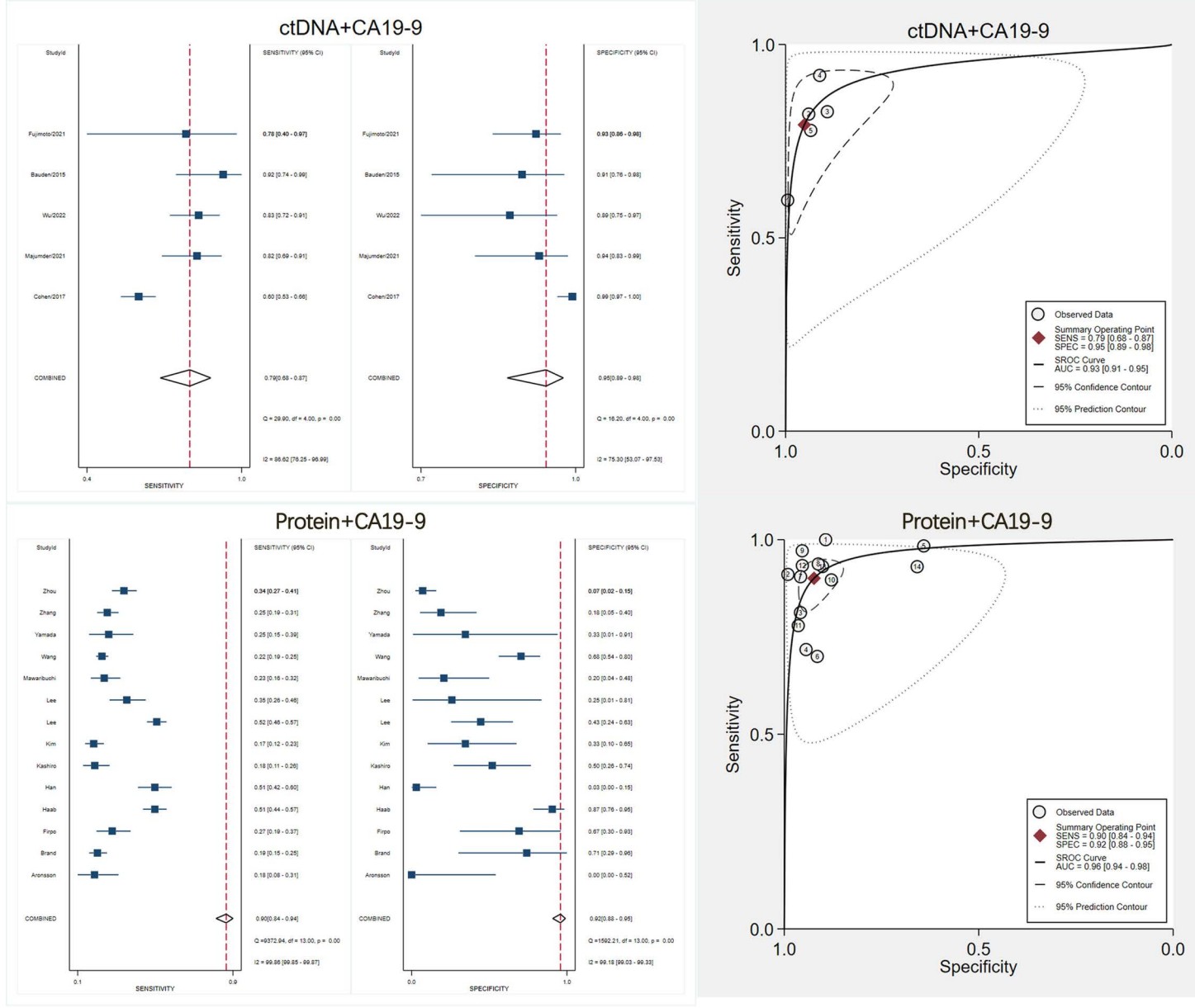

**Fig 7. Combined Diagnostic Performance of CA19-9 with Protein Biomarkers and ctDNA for Early Pancreatic Cancer Detection.**

within each biomarker type to improve diagnostic accuracy and consistency. Furthermore, exploring the combined use of biomarkers and integrating them with imaging techniques could enhance diagnostic performance. Finally, investigating the cost-effectiveness and feasibility of routine screening, especially for high-risk populations, is essential for practical clinical application.

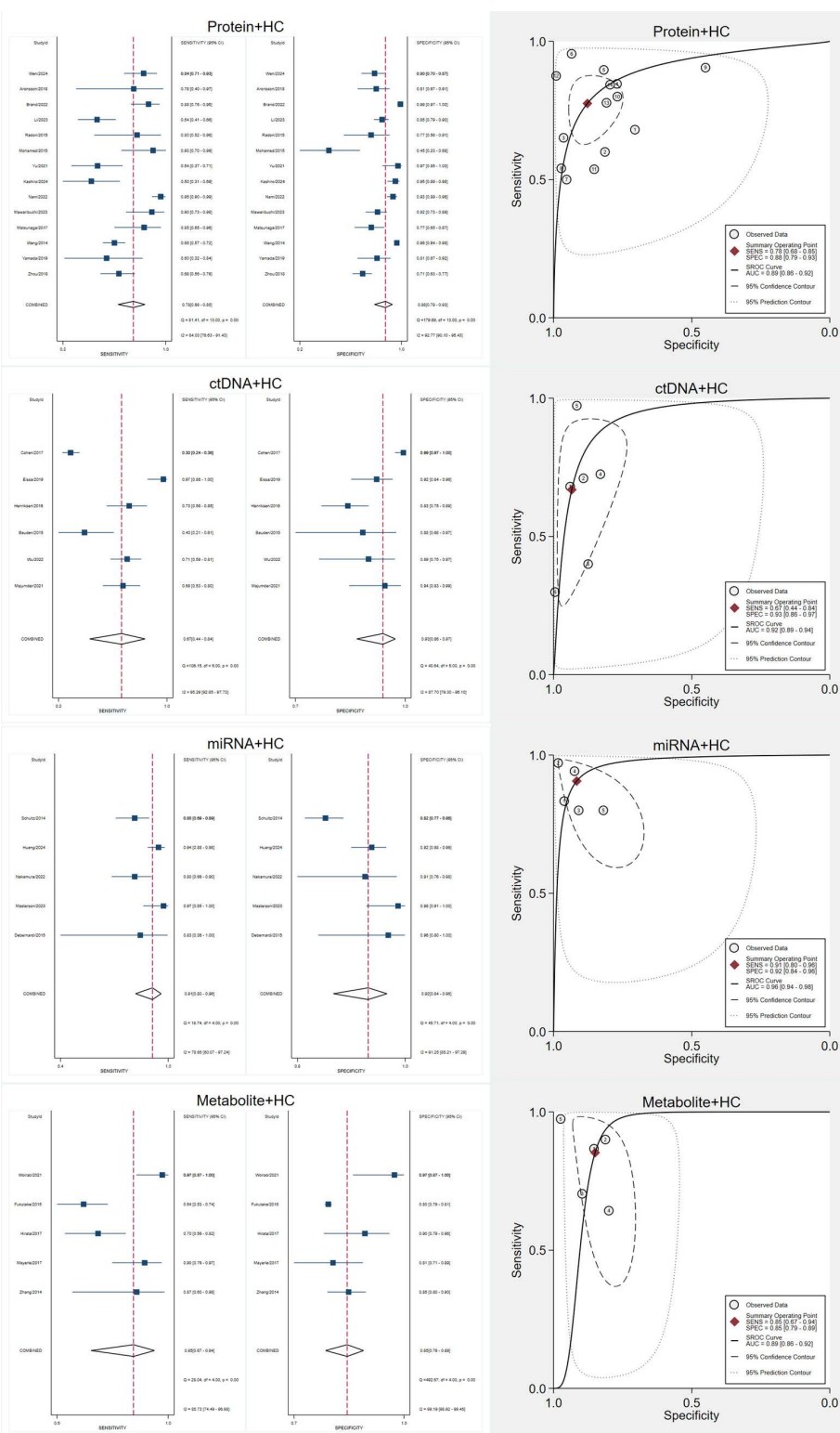

**Fig 8. Diagnostic Performance of Biomarkers in Healthy Control Populations for Early Pancreatic Cancer Detection.**

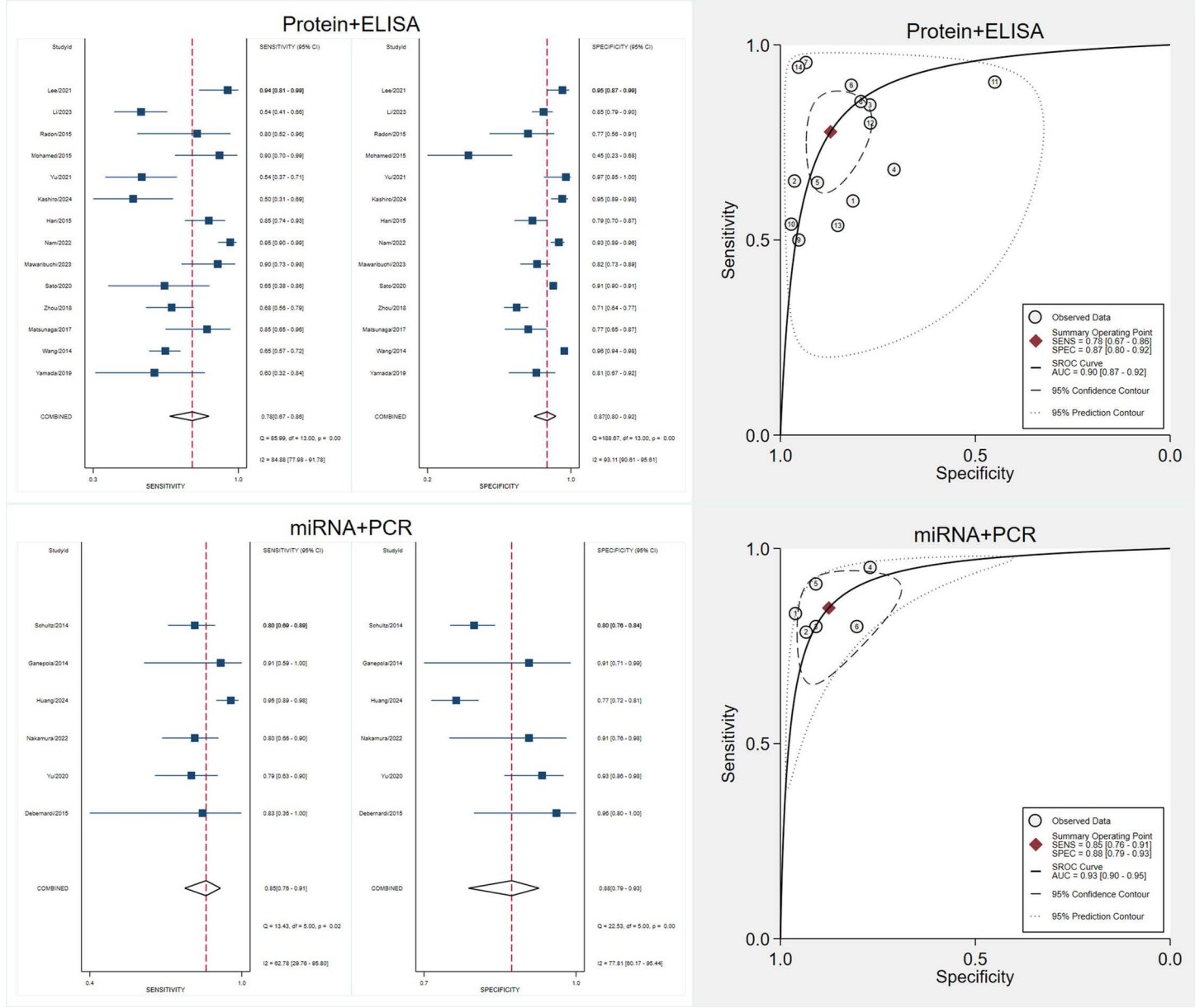

**Fig 9. Diagnostic Accuracy of Protein Biomarkers and miRNAs Using Specific Detection Methods for Early Pancreatic Cancer.**

## Conclusion

This meta-analysis highlights the promising diagnostic performance of novel biomarkers, particularly miRNA and protein biomarkers, in early pancreatic cancer detection. Despite challenges from heterogeneity and methodological differences, these biomarkers offer potential for improving early diagnosis and patient prognosis. Further investigation through well-designed prospective studies is warranted to integrate these biomarkers into clinical practice.

## Supporting information

**S1 file. Supporting information.** S1 Fig. Sensitivity with CA19–9 + Protein Biomarker (Either-Positive).S2 Fig. Sensitivity with CA19–9 + Protein Biomarker (Predictive/ Machine Learning). S1 Table. Detailed search strategy for the systematic review.S2 Table. QUADAS-2 review items.S3 Table. QUADAS-2 review results.S4 Table. Characteristics and diagnostic outcomes of studies using protein or ctDNA combined with CA19–9 biomarkers.S5 Table. Characteristics and diagnostic outcomes of studies in healthy control populations.S6 Table. Diagnostic accuracy of protein biomarkers and miRNA bio-markers using specific detection methods for early pancreatic cancer.S7 Table. Results of leave-one-out analysis.
(PDF)

**S2 file. Table of all studies.**
(PDF)

**S3 file. Table of data extraction.**
(PDF)

**S4 file. PRISMA Checklist.**
(PDF)

## Acknowledgments

Declaration of personal and funding interests: None.

## Author contributions

**Conceptualization:** ZeYi Zheng.

**Data curation:** ZeYi Zheng, Fei Yan.

**Formal analysis:** ZeYi Zheng, Ziyu Lu, Fei Yan.

**Methodology:** ZeYi Zheng.

**Project administration:** ZeYi Zheng.

**Visualization:** ZeYi Zheng.

**Writing – original draft:** ZeYi Zheng.

**Writing – review & editing:** Ziyu Lu, Yani Song.

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
