## [Decision Letter · Decision Letter 0]

11 Mar 2025

PONE-D-25-03633The Role of Novel Biomarkers in the Early Diagnosis of Pancreatic Cancer: A Systematic Review and Meta-AnalysisPLOS ONE

Dear Dr. Song,

Thank you for submitting your manuscript to PLOS ONE. After careful consideration, we feel that it has merit but does not fully meet PLOS ONE’s publication criteria as it currently stands. Therefore, we invite you to submit a revised version of the manuscript that addresses the points raised during the review process.

We look forward to receiving your revised manuscript.

Kind regards,

Shuai Ren

Academic Editor

PLOS ONE

Journal Requirements:

4. Please include captions for your Supporting Information files at the end of your manuscript, and update any in-text citations to match accordingly. Please see our Supporting Information guidelines for more information: http://journals.plos.org/plosone/s/supporting-information .

5. As required by our policy on Data Availability, please ensure your manuscript or supplementary information includes the following:

Additional Editor Comments:

In the introduction part, the authors should emphasize the dire need of diagnosis pancreatic cancer at an early stage and following references could be added: doi: 10.4251/wjgo.v16.i4.1256; doi: 10.1177/20552076231179007.

There are many other biomarkers useful for diagnosis of early-stage pancreatic cancer, following references could be added such as: doi:10.1002/cam4.5296; doi:10.1177/10732748251316602.

Reviewers' comments:

Reviewer's Responses to Questions

**Comments to the Author**

1. Is the manuscript technically sound, and do the data support the conclusions?

Reviewer #1: Yes

Reviewer #2: No

2. Has the statistical analysis been performed appropriately and rigorously? 

Reviewer #1: Yes

Reviewer #2: Yes

3. Have the authors made all data underlying the findings in their manuscript fully available?

Reviewer #1: Yes

Reviewer #2: No

4. Is the manuscript presented in an intelligible fashion and written in standard English?

Reviewer #1: Yes

Reviewer #2: Yes

5. Review Comments to the Author

Reviewer #1: There are several inconsistencies in the font size and type in the manuscript. Check and fix them.

When you reference a figure or table in-text, do not include the description of that figure or table)

Line 96-113 : I think you could make a table for this or list the points in bullets to make it easier to read through.

Line 104: State the full form of QUADAS-2

Study selection: How did you access interrater reliability?

Line 147-149: What guided setting a benchmark score of 3 for inclusion?

Line 176: “Insufficient sample size” could be regarded as vague. What sample size was considered insufficient and why?

Line 181: “Tables 1 and Table2” fix the typo.

Line 183-186: These sentences are confusing.

Line 211: Kindly list the major proteins identified in these studies.

Line 307: You can include the funnel plot for this finding.

Line 296: Kindly list the major metabolites identified in these studies.

Reviewer #2: Manuscript ID: PONE-D-25-03633

Thank you for the opportunity to review this manuscript. The manuscript provides a comprehensive overview of biomarkers for the early diagnosis of pancreatic cancer. However, several aspects of the methodology require further elaboration.

1. The authors pooled data by biomarker type. However, even within each type, there is variability in the specific biomarkers and their cut-off values. I am not a subject expert, but I wonder how meaningful it is to combine such different biomarkers.

2. The authors state that studies with a QUADAS-2 score of 3 or higher were included. QUADAS-2 has four domains for bias and three for applicability. Could you clarify how the scoring was performed? Does a total score of 7 indicate inclusion, or was the focus only on bias? Since bias and applicability have different implications, combining them does not seem appropriate.

3. Please provide the QUADAS-2 assessment results for each study.

4. Please specify the search dates.

5. The authors report the combination of CA-19 and other biomarkers. Could you elaborate on how these were combined and whether this varied across studies? Was a positive result defined as either marker being positive or both being positive? For example, Yamada (2019) reported the sensitivity and specificity of bombinin anti-3’-sialyllactose IgG and CA-19 using predictive equations rather than a simple combination of the two tests. How was this handled in the analysis? The text provides data for calculating sensitivity based on a simple definition of positivity (resulting in a sensitivity of 93%, as in this review, with a positive result defined as either being positive), but there is no information to calculate the specificity of the combined screening using the same definition. Kashiro et al., on the other hand, appear to have combined two binary results (presumably defining positivity as either test being positive).

6. Please report the full results of the sensitivity analysis in the supplementary material. The manuscript states that sensitivity analyses did not show significant changes, yet the discussion mentions that "sensitivity analyses indicated heterogeneity might be associated with very large or small sample sizes." This seems inconsistent.

7. Exclusion criteria: Please clarify what is meant by "studies with excessive heterogeneity in design, patient population, biomarker detection method." How were these defined? Additionally, what threshold was used for defining "small-scale studies"?

8. Could you report how pancreatic cancer was excluded in each study?

9. The manuscript states as a strength that "advanced statistical models accommodated variability and calculated pooled estimates." This is too general. The analysis appears to use a standard random-effects meta-analysis. Please clarify.

10. The manuscript states that "most studies (n = 27) evaluated more than one biomarker," yet Table 2 reports only a single set of numbers per study. Please clarify this discrepancy.

11. The manuscript states that "all relevant data are within the manuscript and its Supporting Information files," but the numbers of TP, FP, FN, and TN needed to reproduce Figure 7 are missing. Please provide these.

12. In Table 2, what does "NC" stand for in the control group?

6. PLOS authors have the option to publish the peer review history of their article (what does this mean? ). If published, this will include your full peer review and any attached files.

**Do you want your identity to be public for this peer review?** For information about this choice, including consent withdrawal, please see our Privacy Policy .

Reviewer #1: No

Reviewer #2: No

---

## [Author Response · Author response to Decision Letter 0]

25 Mar 2025

Response to Editor

Dear Editor,

Thank you for the opportunity to revise our manuscript. We have carefully revised the manuscript in accordance with the reviewers' suggestions and adjusted the manuscript formatting to match the PLOS ONE template, ensuring it fully complies with the journal's style requirements.

All raw data necessary for reproducing our findings have been provided within the main manuscript and supplementary materials. Additionally, we have included the names of the data extractors (Line 153) and specified the date of data extraction (Line 140) in the main text. A comprehensive table showing the complete risk of bias and quality assessment for each included study is provided in the supplementary material (STable 3).

In response to the editorial requirements, we have also made the following additions to the supplementary files:

A numbered table of all studies identified through the literature search, including those excluded from the analyses along with the reasons for exclusion;

A complete table of all data extracted from the primary research sources for the systematic review and/or meta-analysis, including the names of data extractors, dates of extraction, and confirmation of study eligibility;

Moreover, an explanation of how missing data were handled has been incorporated into the manuscript (Lines 170–174).

Furthermore, we have emphasized the urgent clinical need for early diagnosis of pancreatic cancer in the Introduction section (Lines 69-72) and highlighted the potential utility of numerous novel biomarkers for diagnosing pancreatic cancer at an early stage(Lines 86-94) .

We hope these revisions have sufficiently addressed the reviewers' concerns and improved the quality of our manuscript.

Thank you again for considering our manuscript for publication in PLOS ONE. We look forward to your favorable response.

Yours sincerely,

Yani Song

On behalf of all co-authors 

Response to Reviewer #1

Dear Reviewer #1,

Thank you for your thorough review and valuable comments. We have carefully addressed each point you raised and implemented the suggested revisions to enhance the clarity, consistency, and presentation of our manuscript. Below, please find our point-by-point responses:

1. There are several inconsistencies in the font size and type in the manuscript. Check and fix them.

Response: We appreciate your observation. We have thoroughly reviewed the manuscript and corrected any inconsistencies in font size and type. These adjustments ensure a more uniform and professional appearance throughout the text.

2. When you reference a figure or table in the text, do not include the description of that figure or table.

Response: Thank you for the clarification. We have removed any descriptive text when referencing figures and tables in the main body of the manuscript. Only the figure or table number is now included.

3. Line 96-113: I think you could make a table for this or list the points in bullets to make it easier to read through.

Response: Thank you for this helpful suggestion. In response, we revised the content around lines 96–113 by listing the key points in bullet format (lines 106–124). We believe this revision improves readability and clarity for the reader.

4. Line 104: State the full form of QUADAS-2

Study selection: How did you access interrater reliability?

Response: We have now stated the full form of QUADAS-2(Line 114-115) and explained each of its domains in greater detail (Supplementary Tables 2 and 3). Additionally, we clarified how we applied this tool to assess risk of bias and applicability concerns in the included studies. We clarified that two independent reviewers assessed each study using the QUADAS-2 tool. Disagreements were resolved through discussion or consultation with a third reviewer. (lines 177–186)

5. Line 147-149: What guided setting a benchmark score of 3 for inclusion?

Response: We have clarified the rationale for using a QUADAS-2 threshold. In our revised text (lines 177–186), we explain that studies with high risk of bias in at most one domain (out of four bias domains) were included. The original reference to a “3-point” standard has been removed to avoid confusion. Instead, we specify that at least three domains of bias must not be rated as high risk for a study to be included.

6. Line 176: “Insufficient sample size” could be regarded as vague. What sample size was considered insufficient and why?

Response: We have replaced the phrase “insufficient sample size” with a clear cutoff of fewer than 30 participants (line 212-213). Studies with fewer than 30 participants were excluded due to concerns about statistical power and reliable estimation of diagnostic performance.

7. Line 181: “Tables 1 and Table2” fix the typo.

Response: We have corrected this typographical error in line 219.

8. Line 183-186: These sentences are confusing.

Response: We have revised the text in lines 221–227. We now clearly distinguish between healthy control group (HC) and non-cancer control group(NC) including pancreatitis, benign pancreatic diseases in the included studies.

9. Line 211: Kindly list the major proteins identified in these studies.

Response: We have expanded Table 3 to include the specific protein biomarkers evaluated in each study (line 245). Each biomarker is now clearly listed, providing detailed insight into the diagnostic markers investigated.

10. Line 307: You can include the funnel plot for this finding.

Response: We have included a funnel plot (Fig 6.E) for the results related to potential publication bias (lines 370–371). This funnel plot was generated using Deeks’ test (p = 0.04), and we note in the Discussion that any relevant findings should be interpreted with caution due to potential bias.

11. Line 296: Kindly list the major metabolites identified in these studies.

Response: As with the proteins, we have updated Table 3 to include all major metabolites reported in the relevant studies (line 245).

We sincerely appreciate your detailed review and constructive feedback. Your suggestions have significantly enhanced the clarity, coherence, and scientific rigor of our manuscript. We hope our revisions meet your expectations. Please let us know if there are any additional concerns or suggestions.

Kind regards,

Yani Song

Response to Reviewer #2

Dear Reviewer #2,

Thank you for your thoughtful review and for recognizing the comprehensive scope of our work on early diagnostic biomarkers for pancreatic cancer. We value your insightful comments on the methodology and have addressed each point in detail below:

1. The authors pooled data by biomarker type. However, even within each type, there is variability in the specific biomarkers and their cut-off values. I am not a subject expert, but I wonder how meaningful it is to combine such different biomarkers.

Response: We sincerely appreciate the reviewer’s insightful observation regarding the heterogeneity within biomarker categories. This is a critical consideration, and we acknowledge that pooling studies with distinct biomarkers and varying thresholds introduces potential limitations. Below, we clarify our rationale and address the validity of this approach:

(1). Rationale for Pooling by Biomarker Type

Our primary goal was to evaluate the overall diagnostic potential of broad biomarker categories (e.g., miRNAs, proteins) for early pancreatic cancer detection. While individual biomarkers within a category may differ mechanistically, grouping them by type allows us to:

(1.1). Assess the overall performance of biomarker categories (e.g., miRNAs vs. proteins), guiding future research priorities.

(1.2). Compare liquid biopsy-based biomarker classes (e.g., ctDNA vs. metabolites) to determine if they offer distinct advantages in sensitivity or specificity.

(1.3). Provide clinicians with actionable insights on which biomarker categories might complement existing diagnostic tools like CA19-9.

(2).Addressing Heterogeneity Within Categories

We agree that variability in biomarkers and their thresholds can affect pooled estimates. To mitigate this, We used a random-effects model, which accounts for heterogeneity by assuming true effect sizes vary across studies. Subgroup analyses were performed, such as separating studies using ELISA for protein biomarkers from those using mass spectrometry (Fig. 9). These analyses revealed no significant differences in performance, suggesting that methodological variability had limited impact. We explicitly reported high I² values (e.g., 83-98% for proteins and metabolites), emphasizing the need for cautious interpretation.

(3).Clinical and Biological Justification

Despite variability, biomarkers within a class often share biological relevance:

miRNAs: Many regulate overlapping oncogenic pathways and are consistently dysregulated in pancreatic cancer.

Proteins: While individual markers vary (e.g., S100P, MIC-1), they often reflect tumor-associated inflammation or immune evasion.

Metabolites: Altered lipid or amino acid profiles broadly indicate metabolic reprogramming, a hallmark of cancer.

Pooling these biomarkers provides a holistic view of their collective diagnostic utility, even if individual candidates require further validation.

(4).Limitations and Future Directions

We recognize the limitations of this approach and have revised the Discussion to emphasize:

The need for standardized biomarker panels: Future studies should prioritize validating specific combinations rather than isolated markers.

The importance of harmonizing thresholds: Cut-off values should be optimized in large cohorts to reduce variability. We acknowledge that individual biomarkers may exhibit different cut-off values, sensitivities, and specificities, which could limit the generalizability of pooled estimates.

While pooling biomarkers by category introduces heterogeneity, this approach aligns with the study’s exploratory aim to compare broad diagnostic strategies. We agree that future research should focus on standardizing specific biomarker panels, and we thank the reviewer for raising this important point.

We further have emphasized this limitation in our Discussion. (Lines 527-538)

2. The authors state that studies with a QUADAS-2 score of 3 or higher were included. QUADAS-2 has four domains for bias and three for applicability. Could you clarify how the scoring was performed? Does a total score of 7 indicate inclusion, or was the focus only on bias? Since bias and applicability have different implications, combining them does not seem appropriate.

Response: We agree with your concern that combining bias and applicability domains is inappropriate. In our revised manuscript (lines 177–186), we clarify that we did not merge these domains into a single score. Instead, we focused primarily on the four bias domains. Studies with at most one domain rated as high risk of bias in these four areas were included. We have removed references to a “3-point” system and now explain precisely how we assessed and reported each domain’s outcome. The results are provided in Supplementary Table 3.

3. Please provide the QUADAS-2 assessment results for each study.

Response: We have now included the full QUADAS-2 evaluation for each study in Supplementary Table 3.

4. Please specify the search dates.

Response: We have added the exact date of our literature search (June 1, 2024) to the Methods section (line 140), ensuring greater transparency regarding our search strategy.

5. The authors report the combination of CA-19 and other biomarkers. Could you elaborate on how these were combined and whether this varied across studies? Was a positive result defined as either marker being positive or both being positive? For example, Yamada (2019) reported the sensitivity and specificity of bombinin anti-3’-sialyllactose IgG and CA-19 using predictive equations rather than a simple combination of the two tests. How was this handled in the analysis? The text provides data for calculating sensitivity based on a simple definition of positivity (resulting in a sensitivity of 93%, as in this review, with a positive result defined as either being positive), but there is no information to calculate the specificity of the combined screening using the same definition. Kashiro et al., on the other hand, appear to have combined two binary results (presumably defining positivity as either test being positive).

Response: Thank you for raising this important issue. To address your concern, we have carefully re-reviewed and re-classified the included studies concerning the combined analysis of CA19-9 with other biomarkers. Based on the methods reported in these studies, we categorized the combined analyses into two approaches:

Either-positive principle:

In these studies, sensitivity was calculated such that if either CA19-9 or the novel biomarker was positive, the overall test was considered positive. Conversely, specificity was calculated by requiring both CA19-9 and the novel biomarker to be negative to classify a case as negative. In other words, this approach applied an “OR” principle for positivity and an “AND” principle for negativity.

Predictive modeling or machine-learning methods:

Another group of studies utilized regression equations, scoring systems, or machine-learning algorithms (such as logistic regression, multi-marker panels, or other algorithms) to estimate the sensitivity and specificity of combined testing. Typically, an integrated score or predicted probability was used as a threshold: results exceeding this threshold were classified as positive, while those below were classified as negative. Hence, sensitivity and specificity in these studies were not calculated by simple binary combinations but rather determined directly through predictive modeling or receiver operating characteristic (ROC) analysis.

For these studies, we first conducted an overall meta-analysis to estimate the diagnostic performance of biomarkers combined with CA19-9. Subsequently, subgroup analyses were conducted according to the different combination methods, and we assessed whether significant differences existed at the subgroup level.

Relevant methodological clarifications, along with the results obtained and their implications in the discussion, have been added to the manuscript.

We sincerely appreciate your highlighting this matter, which has greatly contributed to enhancing the clarity and rigor of our manuscript. Thank you again for your constructive feedback (Lines 396-431; Lines 490-504; Supplementary Figs 1 and 2).

6. Please report the full results of the sensitivity analysis in the supplementary material. The manuscript states that sensitivity analyses did not show significant changes, yet the discussion mentions that "sensitivity analyses indicated heterogeneity might be associated with very large or small sample sizes." This seems inconsistent.

Response: We have reviewed and consolidated our sensitivity analysis findings. All results, including leave-one-out analysis, have been added to the Supplementary Material (Supplementary Fig 7). We clarified that while small or large sample sizes could theoretically introduce heterogeneity, the actual sensitivity analyses did not reveal significant changes (lines 523–524). We thank you for pointing out this discrepancy, which we have now resolved.

7. Exclusion criteria: Please clarify what is meant by "studies with excessive heterogeneity in design, patient population, biomarker detection method." How were these defined? Additionally, what threshold was used for defining "small-scale studies"?

Response: We have clarified that “excessive heterogeneity” refers to high risk of bias or inapplicability in multiple QUADAS-2 domains, leading to exclusion. Furthermore, we now explicitly define “small-scale studies” as those enrolling fewer than 30 participants, as they lack sufficient statistical power for reliable conclusions. (lines 121–124)

8. Could you report how pancreatic cancer was excluded in each study?

Response: We have added a detailed explanation in the Methods section, clarifying how each included study ensured that enrolled participants in the control arm did not have pancreatic cancer. (lines 125–137)

9.

---

## [Editor Report · Decision Letter 1]

26 Mar 2025

The Role of Novel Biomarkers in the Early Diagnosis of Pancreatic Cancer: A Systematic Review and Meta-Analysis

PONE-D-25-03633R1

Dear Dr. Song,

We’re pleased to inform you that your manuscript has been judged scientifically suitable for publication and will be formally accepted for publication once it meets all outstanding technical requirements.

Kind regards,

Shuai Ren

Academic Editor

PLOS ONE

Additional Editor Comments (optional):

Congratulations to all the authors and thank you for addressing all comments and suggestions during the review process.
---

## [Editor Report · Acceptance letter]

PONE-D-25-03633R1

PLOS ONE

Dear Dr. Song,

I'm pleased to inform you that your manuscript has been deemed suitable for publication in PLOS ONE. Congratulations! Your manuscript is now being handed over to our production team.

Kind regards,

on behalf of

Dr. Shuai Ren

Academic Editor

PLOS ONE